# Simultaneous ventilation in the Covid-19 pandemic. A bench study

Claude Guérin[1,2,3]*, Martin Cour[1,2], Neven Stevic[1], Florian Degivry[1], Erwan L'Her[4,5], Bruno Louis[3], Laurent Argaud[1,2]

1 Médecine Intensive-Réanimation, Hospices Civils de Lyon, Groupement Hospitalier Centre, Hôpital Edouard Herriot, Lyon, France, 2 Faculté de Médecine Lyon-Est, Université de Lyon, Lyon, France, 3 Institut Mondor de Recherches Biomédicales, INSERM UMR 955 Eq13—CNRS ERL 7000, Créteil, France, 4 Médecine Intensive-Réanimation, CHU de Brest, Brest, France, 5 LATIM Inserm UMR 1101, Université de Brest, Brest, France

☯ These authors contributed equally to this work.
* claude.guerin@chu-lyon.fr

**Data Availability Statement:** All relevant data are within the manuscript.

**Funding:** The author(s) received no specific funding for this work.

## Abstract

COVID-19 pandemic sets the healthcare system to a shortage of ventilators. We aimed at assessing tidal volume ($V_T$) delivery and air recirculation during expiration when one ventilator is divided into 2 test-lungs. The study was performed in a research laboratory in a medical ICU of a University hospital. An ICU (V500) and a lower-level ventilator (Elisée 350) were attached to two test-lungs (QuickLung) through a dedicated flow-splitter. A 50 mL/cmH$_2$O Compliance (C) and 5 cmH$_2$O/L/s Resistance (R) were set in both A and B test-lungs (A C50R5 / B C50R5, step1), A C50-R20 / B C20-R20 (step 2), A C20-R20 / B C10-R20 (step 3), and A C50-R20 / B C20-R5 (step 4). Each ventilator was set in volume and pressure control mode to deliver 800mL $V_T$. We assessed $V_T$ from a pneumotachograph placed immediately before each lung, pendelluft air, and expiratory resistance (circuit and valve). Values are median (1$^{st}$-3$^{rd}$ quartiles) and compared between ventilators by non-parametric tests. Between Elisée 350 and V500 in volume control $V_T$ in A/B test-lungs were 381/387 vs. 412/433 mL in step 1, 501/270 vs. 492/370 mL in step 2, 509/237 vs. 496/332 mL in step 3, and 496/281 vs. 480/329 mL in step 4. In pressure control the corresponding values were 373/336 vs. 430/414 mL, 416/185 vs. 322/234 mL, 193/108 vs. 176/ 92 mL and 422/201 vs. 481/329mL, respectively (P<0.001 between ventilators at each step for each volume). Pendelluft air volume ranged between 0.7 to 37.8 ml and negatively correlated with expiratory resistance in steps 2 and 3. The lower-level ventilator performed closely to the ICU ventilator. In the clinical setting, these findings suggest that, due to dependence of $V_T$ to C, pressure control should be preferred to maintain adequate $V_T$ at least in one patient when C and/or R changes abruptly and monitoring of $V_T$ should be done carefully. Increasing expiratory resistance should reduce pendelluft volume.

## Introduction

During the COVID-19 pandemic, a risk of a shortage of ICU ventilators was claimed very early [1]. As the poliomyelitis pandemic prompted the caregivers to discover tracheotomy,

**Competing interests:** The authors have declared that no competing interests exist.

iron lung, and mechanical ventilation, the current COVID-19 pandemic prompted innovative solutions [2]. They include ventilator multipliers [3], portable and open-source designs of ventilators [4], and frugal ventilators [5]. Simultaneous ventilation provides ventilatory support to two or more patients with the same ventilator [6]. This approach raised ethical issues [7] due to the many technical problems to solve from sharing the same ventilator with patients with different respiratory mechanics and, hence different requirements [8]. For simultaneous ventilation, with no means of independently controlling positive end-expiratory pressure (PEEP) and tidal volume ($V_T$), patients sharing the same ventilator should have respiratory mechanics as similar as possible. In this case, in volume control ventilation and pressure control ventilation mode, each patient is expected to receive half of the set $V_T$. Any decrease in compliance and/or increase in resistance in one patient will decrease $V_T$ in each mode [9, 10]. For the other patient with unchanged compliance and resistance, $V_T$ will depend, at least in part, on mechanical ventilation mode. The degree to which differences in mechanics cause differences in $V_T$ depends both on how the mechanics differ and on the mode of ventilation (i.e. volume versus pressure control). In general, on test bench, changes in one test-lung cause changes in the other(s) test-lung(s), regardless of the model [9]. Moreover, air may recirculate during inspiration and expiration from one test-lung to the other (pendelluft air). This issue exposes patients to the risk of $CO_2$ retention and cross-transmission of infection. The test-lung with the shortest inspiratory time constant, i.e. with the lowest product of resistance by compliance, breathed out faster (earlier) while the other was still filling in. However, the role of expiratory resistance (circuit and ventilator valve) on pendelluft air has not been previously addressed. Nevertheless, the feasibility and safety of simultaneous ventilation have been reported recently in a few patients highly selected, deeply monitored, and for a few hours [11–13].

Because simultaneous ventilation is still experimental and not completely investigated, we designed a bench study where 2 or 3 test-lungs with different respiratory mechanics were attached to the same ventilator. We compared a high-performance ICU ventilator and a ventilator used for patient transportation. We assessed $V_T$ delivery, pendelluft air volume, and expiratory resistance.

## Methods

Two ventilators were tested: the Elisée 350 (ResMed, Saint-Priest, France), which is a turbine-driven ventilator used for patient transportation, and in stepdown-units and the V500 ICU ventilator (Draeger, Luebeck, Germany). They were attached to two (A and B) test-lungs (QuickLung test, IngMar Medical, Inc., Pittsburgh, PA) equipped with resistance (R) of 5, 20, and 50 cmH$_2$O/L/s and compliance (C) of 10, 20, and 50 mL/cmH$_2$O in the first part (Part I) of the experiment. In the second part (Part II) a third test-lung (SelfTestLung, Draeger, Luebeck, Germany) of 10 mL/cmH$_2$O compliance and 20 cmH$_2$O/L/s resistance was added (Table 1). We calibrated the ventilator and checked the circuit leakage before starting the experiments. Both ventilators compensated for circuit compliance. The same standard double-limb ventilator circuit of 22 mm internal diameter (Intersurgical, Fontenay sous bois, France) was used in both ventilators and for each experiment. The calibration process was done with this circuit attached to the two and three test-lungs designs.

Besides, high-efficiency particulate air filter (HEPA Isogard, Gibeck, Indianapolis, IN) in front of each test-lung and specific flow-splitters (MICHELIN Molding Solutions, Michelin, Clermont-Ferrand, France) were used (Fig 1). Airflow was measured by pneumotachographs (Hamilton, Sidam, Mirandola, Italy) and airway pressure (Paw) by pressure transducers (Gabarith PMSET 1DT-XX, Becton Dickinson, Singapore).

**Table 1. Study design.**

| Parts of the experiments | Steps | Test-lung A | τ test-lung A | Test-lung B | τ test-lung B | Test-lung C | τ test-lung C |
|---|---|---|---|---|---|---|---|
| | | QuickTest lung | | QuickTest lung | | SelfTestLung | |
| Part I. Two test-lungs design | 1 | C50-R5 | 0.25 | C50-R5 | 0.25 | none | |
| | 2 | C50-R20 | 1.00 | C20-R20 | 0.40 | none | |
| | 3 | C20-R20 | 0.40 | C10-R20 | 0.20 | none | |
| | 4 | C50-R20 | 1.00 | C20-R5 | 0.10 | none | |
| Part II. Three test-lungs design | 1 | C50-R5 | 0.25 | C50-R5 | 0.25 | C10-R20 | 0.20 |
| | 2 | C50-R20 | 1.00 | C20-R20 | 0.40 | C10-R20 | 0.20 |
| | 3 | C20-R20 | 0.40 | C10-R20 | 0.20 | C10-R20 | 0.20 |
| | 4 | C50-R20 | 1.00 | C20-R5 | 0.10 | C10-R20 | 0.20 |

C: compliance in ml/cmH$_2$O, R: resistance in cmH$_2$O/L/s, τ: time constant, expressed in seconds.

Two designs were used for each ventilator in both volume and pressure control ventilation (Table 1).

In the first part, the two QuickLung models were arranged in parallel (Fig 1) and 4 steps performed. In step 1, a similar C50-R5 was applied to both test-lungs. The 3 other steps are detailed in Table 1. In volume control, $V_T$ (A squared flow profile) was set to 800 mL to deliver 400 mL to each test-lung. PEEP was set to 15 cmH$_2$O, respiratory rate to 20 breaths/min, inspiratory time to 0.90 s, inspiratory pause to 0.1 s, inspiratory to expiratory ratio to 1:2, inspiratory flow to 60 L/min, inspired oxygen fraction ($F_IO_2$) to 21%.

In pressure control, the driving pressure and inspiratory time were set to get a $V_T$ of 800 mL in step1. The above settings were kept for the other steps because the steps of the experiment (Table 1) were thought to reflect the time course of asymmetrical disturbance that may occur between patients attached to the same ventilator, i.e. sudden loss of lung volume in one patient (pneumothorax, atelectasis), and hence the ventilator has no reason to be set differently before the situation occurs. Such a double circuit design is also suitable at the time of patient selection for simultaneous ventilation. The safety guideline is to share the single ventilator between patients with respiratory mechanics as close as possible [8, 14]. We wanted to explore how much $V_T$ would depart from each patient to the other when the respiratory mechanics markedly differ between them.

In the second design, a third test-lung with a fixed C20-R20 was added in parallel to the previous C-R (Table 1) of the two QuickLung models and $V_T$ set to 1000 mL. This part of the

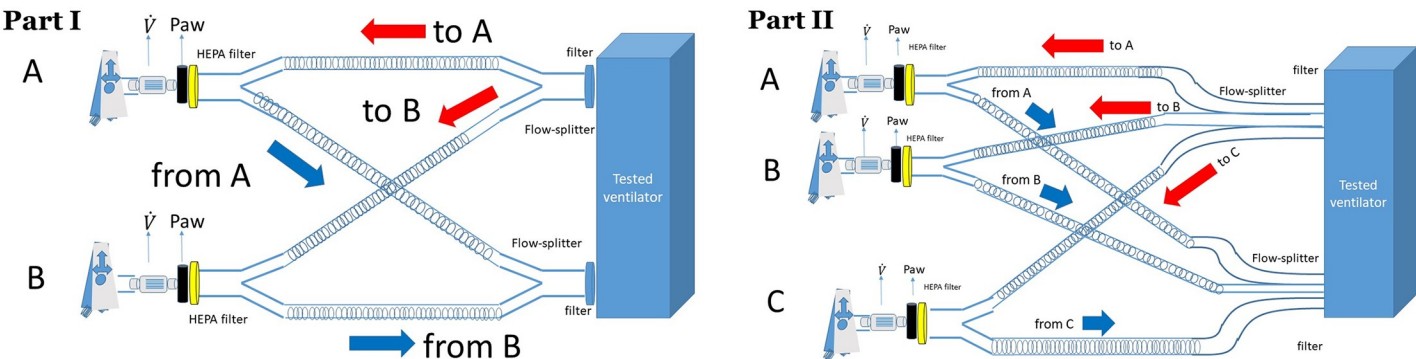

**Fig 1.** Experimental set-up with two test-lungs A and B (Part I) and three test-lungs (Part II). Paw: airway pressure, HEPA: high-efficiency particulate air filter, $\dot{V}$: flow. Red and blue arrows indicate inspiration and expiration, respectively.

study aimed at stretching the asymmetry between patients. Our choice of C-R was in line with the standards [15].

Flow and Paw analog signals were sent to a datalogger (Biopac MP150, Biopac Inc., Goletta, CA). Mechanical ventilation was stabilized for one minute and, then signals were collected for one minute at a 200 Hz sampling rate.

To get a better look at the feasibility of simultaneous ventilation, we recorded some anthropometric data, as well as ventilator settings and respiratory mechanics at 1 hour after start of invasive mechanical ventilation in 20 consecutive patients with COVID-19 acute respiratory distress syndrome (ARDS) hospitalized in our 26-bed medical ICU. This part of the study was approved by our institutional ethics committee (*Comité d'Ethique du CHU de Lyon*) with a waiver for written inform consent because of its retrospective nature of the study.

## Data analysis

Collected data were analyzed off-line by using in-house software specifically developed for the present study (Matlab R2019b, MathWorks, Inc.). $V_T$ was obtained by integration of the flow signal over the inflation time in each test-lung, which can be different between each test-lung (see below) and different from the machine inflation time. Taking care of this is important in a design, like our present one, that accommodates different time constants between test-lungs. The pendelluft air volume was computed as the amount of air that flowed from one test-lung to the other(s) during inspiration and expiration (Fig 2).

The test-lung B with the shortest inspiratory time constant ends up inspiration earlier than test-lung A. At the time of test-lung B ends inspiration, $\dot{V}_B = 0$, and $\dot{V}_A = \dot{V}$. The tidal volume received by test-lung B is equal to the compliance of the test-lung B times maximal Paw (Pmax) minus resistive pressure (PresB), which equal to zero because $\dot{V}_B = 0$. Therefore Pmax equals alveolar pressure in test-lung B (PalvB). At the same time since $\dot{V}_A$ is >0, PalvA is equal to Pmax minus PresA, and hence is lower than PalvB.

At the time of the ventilator ends insufflation, $\dot{V} = 0$, and $\dot{V}_A = -\dot{V}_B$. PalvA is equal to Paw minus PresA (which is the product of resistance through lung A to $\dot{V}_A$ and similarly PalvB is equal to Paw plus PresB (which is the product of resistance through lung B to $\dot{V}_A$). It comes that PalvB>Paw>PalvA.

At the time of test-lung A ends inspiration, $\dot{V}_A = 0$, $\dot{V}_B = \dot{V}$, *and* $\dot{V} < 0$. Therefore, PalvA = Paw and PalvB is equal to Paw minus PresB (which is the product of the resistance through lung B to $\dot{V}$). PalvB is then greater than Paw, which is equal to PalvA.

The hatched area in the lower graph indicates the amount of pendellfut air from patient B to patient A. The Table 2 summarizes the conditions making rebreathing to or not to happen and its computation.

On each breath, the instantaneous expiratory resistance was determined as the ratio of the pressure drop between Paw and PEEP 15 to the corresponding flow at every single sampled data point as previously described [16]. For this computation, we discarded flows lower than 0.01 L/s to avoid extreme values corresponding to the closing of the valve. Therefore, the instantaneous expiratory resistance was determined in roughly 400 instances in each breath. We used the minimal value of the instantaneous resistance in each condition.

The primary end-point was the value of $V_T$ and secondary end-points were pendelluft air volume and minimal instantaneous expiratory resistance. Normal distribution and homogeneity of variables were assessed. For each C-R condition and ventilator, 20 repeated measurements were performed. The values are presented as median (1st-3rd quartiles) and compared across ventilators and each C-R condition by using a parametric or a non-parametric test as

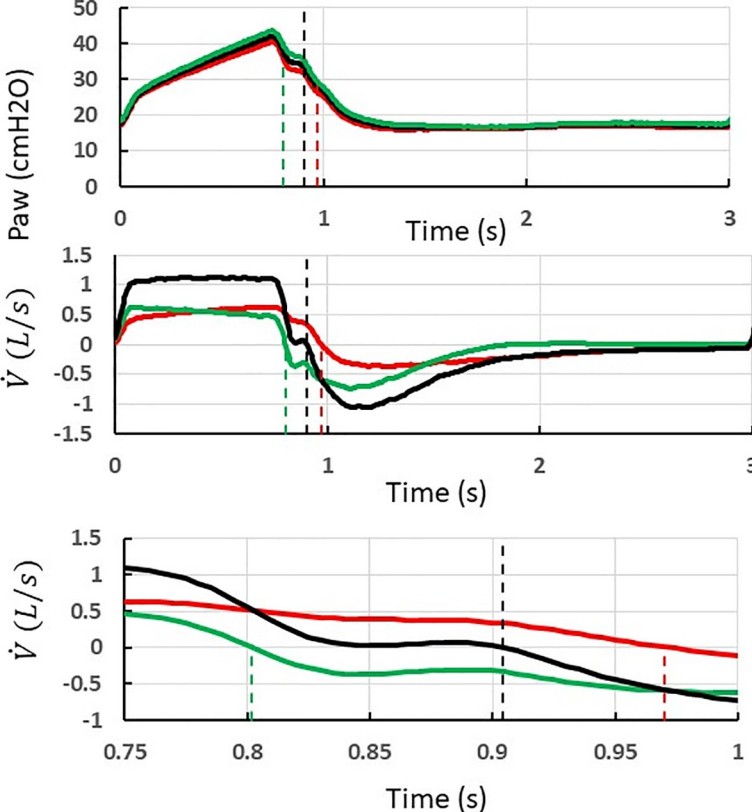

**Fig 2. Method to measure the pendelluft air volume during inspiration and expiration in case of a double circuit and two test-lungs with uneven compliance and resistance ventilated (volume control, step 2, V500 ventilator).** Recording of airway pressure (Paw, top) and flow $\dot{V}$, *middle and bottom*) leaving the ventilator during a breathing cycle. Paw and $\dot{V}$ are distributed in parallel to the test-lung A (red) ($Paw_A$ and $\dot{V}_A$) and the test-lung B (green) ($Paw_B$ and $\dot{V}_B$). The vertical broken lines indicate the corresponding ends of insufflation for the ventilator and each test-lung. The lower panel is a magnification of the end of insufflation.

requested. The statistically significant level was set to P-value < 0.05. The analysis was performed by using R software Version 3.5.2 (R: A Language and Environment for Statistical Computing, R Core Team, R Foundation for Statistical Computing, Vienna, Austria, 2018).

## Results

### V_T delivery

As expected for the step 1, $V_T$ was equally delivered to test-lungs A and B in both volume and pressure control modes (Fig 3). Between Elisée 350 and V500 ventilators with the double

**Table 2. Summary of pendelluft occurrence.**

| $\dot{V}_A$ | $\dot{V}_B$ | Inspiration $\dot{V} > 0$ | Expiration $\dot{V} < 0$ |
|---|---|---|---|
| >0 | >0 | no pendelluft | No pendelluft if $\dot{V}_A$ and $\dot{V}_B < 0$ |
| >0 | <0 | pendelluft from B to A = $\dot{V}_A - \dot{V}$ | pendelluft from B to A = $\dot{V}_A$ |
| <0 | >0 | pendelluft from A to B = $\dot{V}_B - \dot{V}$ | pendelluft from A to B = $\dot{V}_B$ |

$\dot{V}_A$, $\dot{V}_B$: flow to test-lungs A and B, respectively, in Fig 2, $\dot{V}$ flow leaving the ventilator.

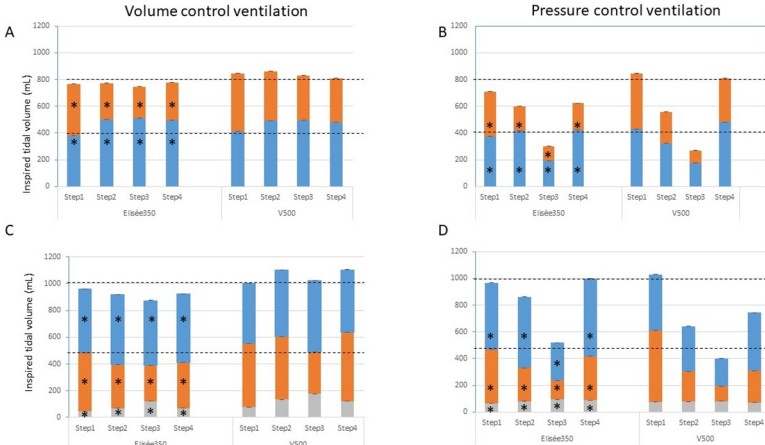

**Fig 3. Inspired tidal volume.** Stacked plots of inspired tidal volume in test-lung A (blue), B (orange), and C (grey) in volume and pressure control ventilation during the 4 steps for each ventilator in the design with two (panels A and B) and three test-lungs (panels C and D). Step1 = C50-R5 for each test-lung, Step2 = A C50-R20 and B C20-R20, Step3 = A C20-R20 and B C10-R20, Step4 = A C50-R20 and B C20-R5. C 10-R20 for test-lung C at each step. Bars are median and the interquartile range (3rd minus 1st quartile) are indicated above bars. The horizontal broken black lines indicate the target tidal volume the ventilator should deliver to each test-lung. *P<0.05 as compared to V500.

circuit, $V_T$ amounted to 381 (379–382) mL in lung A and 387 (385–389) mL in lung B and 412 (409–413) mL and 433 (432–435) mL, respectively (P<0.05 between ventilators) in volume control, and 373 (372–375) mL in lung A and 336 (336–338) mL in lung B vs. 430 (429–430) and 414 (413–416) mL, respectively in pressure control (P<0.05 between ventilators). Both ventilators accurately delivered the targeted value of $V_T$ within 10% boundaries, with Elisée 350 under-delivering $V_T$ by 6 (5–7)% and V500 over-delivering $V_T$ by 4 (2–7)%.

In step 2, compliance was 2.5 times greater in test-lung A than in test-lung B while R was similar in both of them but 4 times greater than in step 1. In volume control, $V_T$ was greater in test-lung A than in test-lung B by a factor of 1.9 with Elisée 350 and 1.33 with V500 (Fig 3). It was 501 (500–503) mL in test-lung A and 270 (269–271) mL in test-lung B with Elisée and 492 (492–493) mL and 370 (360–381) mL, respectively (P<0.001 between ventilators). Therefore, $V_T$ delivery was 20% greater in test-lung A and 33% lower in test-lung B than expected with Elisée, these values being of 23% and 8%, respectively, with V500. A similar figure was observed in step 3 where compliance was twice greater in test-lung A than in test-lung B but 2.5 and 5 times lower, respectively, as compared to step 1, and R was similar in both test-lungs and similar to the one set in step 3 (Fig 3). The same was true for step 4 (Fig 2). Contrary to step 1, in steps 2–4 $V_T$ to test-lung A was greater with Elisée 350 than with V500 and the opposite was true for $V_T$ to test-lung B (Fig 3).

In pressure control with the asymmetrical design, $V_T$ in a given test-lung changed as a result of both the overall $V_T$ decrease due to the greater impedance and the difference between time constants. In step 2, $V_T$ was greater in test-lung A than in test-lung B by a factor of 2.2 with Elisée 350 and of 1.4 with V500 (Fig 3). However, test-lung A accurately received the target $V_T$ (+4%), whilst test-lung B had $V_T$ reduced by 54% as compared to the target $V_T$ with Elisée 350. By contrast, with V500 in test-lung A, $V_T$ was under-delivered by 20% as compared to the target $V_T$, whilst test-lung B received $V_T$ reduced by 43% from the target $V_T$ (P<0.001 between ventilators). The same picture was observed in steps 3 and 4.

In the triple circuit, $V_T$ delivered to test-lung C in volume control was very small according to its low compliance, with lower values with Elisée 350 than V500 (Fig 3C). In step 1, $V_T$ to

test-lung C was 44 (44–45) and 76 (76–76) mL with Elisée 350 and V500, respectively (P<0.001 between ventilators), representing 5 and 8% of whole delivered $V_T$. In the asymmetrical steps 2–4, in particular with V500, $V_T$ to test-lung C increased up to 134 (133–136) mL in step 2, 176 (175–178) mL in step 3, and 122 (121–123) mL in step 4, which sets this stiff lung to the risk of overdistension. In pressure control, $V_T$ to test-lung C was 67 (67–67) mL with Elisée 350 and 75 (74–77) mL with V500 is step 1 (Fig 3D). However, in the asymmetrical 2–4 steps, $V_T$ delivered to that test-lung remained stable and in line with its low compliance preserving it from overdistension (Fig 3D).

## Pendelluft air volume

In the double circuit, the pendelluft air volume ranged from 0.7 to 37.8 mL (Table 3). The direction of the significant differences between ventilators varied across conditions. The amount of pendelluft air volume tended to be lower in pressure control than in volume control. The picture was essentially the same in the triple circuit. The pendelluft air volume to test-lung C was very small, as expected from its very low compliance. The volume of the pendelluft air volume was small as compared to the volume of the expiratory circuit (607 mL).

## Expiratory resistance

The minimal expiratory resistance was different between ventilators in most instances as shown in Fig 4. The differences between ventilators were statistically significant in every comparison without any systematic and consistent direction.

## Lung mechanics in patients with COVID-19 related ARDS

Individual lung mechanical characteristics of 20 successive patients with COVID-19, recorded 1 hour after intubation for ARDS, are shown in Table 4. The median compliance was 35 (28-41) mL/cmH$_2$O and R was 11 (10–14) cmH$_2$O/L/s. Half of patients had a compliance ranging between 30 and 40 mL/cmH$_2$O and only 2 (10%) had a compliance lower than 20 mL/cmH$_2$O.

**Table 3. Pendelluft air volume to each lung.**

| Steps | To test-lung (τ) | Volume control | | Pressure control | | Volume control | | Pressure control | |
|---|---|---|---|---|---|---|---|---|---|
| | | Double Circuit | | | | Triple Circuit | | | |
| | | Elisée 350 | V500 | Elisée 350 | V500 | Elisée 350 | V500 | Elisée 350 | V500 |
| 1 | A (0.25) | 2.3 (1.3–2.9) | 0.3 (0.3–0.3) | 0.4 (0.3–1.8)* | 0.2 (0.2–0.3) | 10.1 (7.5–10.5)* | 0.2 (0.2–0.3) | 3.0 (2.5–3.8)* | 6.0 (5.9–6.0) |
| | B (0.25) | 0.3 (0.2–0.4) | 0.6 (0.6–0.8) | 0.4 (0.3–1.3)* | 0.3 (0.2–0.4) | 0.3 (0.3–0.4)* | 13.9 (13.2–14.1) | 2.9 (2.4–3.3)* | 0.2 (0.2–0.3) |
| | C (0.20) | | | | | 0.0 (0.0–0.1)* | 0.1 (0.1–0.2) | 0.2 (0.1–0.4) | 0.2 (0.2–0.2) |
| 2 | A (1.00) | 14.6 (14.4–14.9)* | 29.6 (15.4–32.1) | 1.9 (1.7–2.2)* | 3.1 (3.0–3.3) | 30.7 (30.2–31.0)* | 52.3 (51.7–53.4) | 7.0 (6.9–7.3) | 7.3 (7.3–7.7) |
| | B (0.40) | 0.7 (0.6–0.7)* | 5.9 (5.6–6.0) | 0.5 (0.5–0.7)* | 4.1 (4.0–4.2) | 0.7 (0.6–0.7)* | 3.7 (3.5–3.9) | 0.5 (0.4–0.5)* | 3.0 (2.8–3.3) |
| | C (0.20) | | | | | 0.2 (0.1–0.2)* | 0.9 (0.81–1.0) | 0.1 (0.1–0.2)* | 0.8 (0.6–0.9) |
| 3 | A (0.40) | 7.0 (6.8–7.3)* | 15.1 (14.5–15.6) | 5.4 (5.0–6.0)* | 0.3 (0.3–0.4) | 18.4 (18.3–18.6)* | 43.4 (43.2–44.3) | 23.0 (22.1–23.6)* | 1.4 (1.3–9.4) |
| | B (0.20) | 0.4 (0.3–0.4)* | 5.3 (4.8–5.8) | 1.6 (1.2–1.9)* | 0.3 (0.2–0.4) | 0.6 (0.5–0.6)* | 1.3 (1.2–1.5) | 4.6 (3.8–5.2)* | 1.2 (1.1–2.2) |
| | C (0.20) | | | | | 0.0 (0.0–0.0)* | 2.0 (1.8–2.4) | 0.1 (0.0–0.2) | 0.2 (0.1–0.2) |
| 4 | A (1.00) | 27.2 (26.7–27.7)* | 37.8 (37.1–38.5) | 2.2 (2.2–2.5)* | 0.2 (0.2–0.3) | 40.4 (40.0–40.7) | 65.3 (64.5–68.0) | 1.6 (1.5–1.9)* | 5.6 (5.4–5.7) |
| | B (0.10) | 0.7 (0.6–0.9) | 1.0 (0.7–1.1) | 0.5 (0.5–0.6)* | 6.7 (6.4–7.1) | 0.8 (0.6–0.9) | 11.8 (11.5–13.2) | 0.7 (0.6–0.8)* | 4.8 (4.6–5.3) |
| | C (0.20) | | | | | 0.1 (0.1–0.2) | 2.6 (2.4–2.5) | 0.1 (0.1–0.2)* | 0.2 (0.2–0.2) |

Values are median (1st-3rd quartiles) in mL except for τ (time constant) in s.

*P<0.05 versus V500.

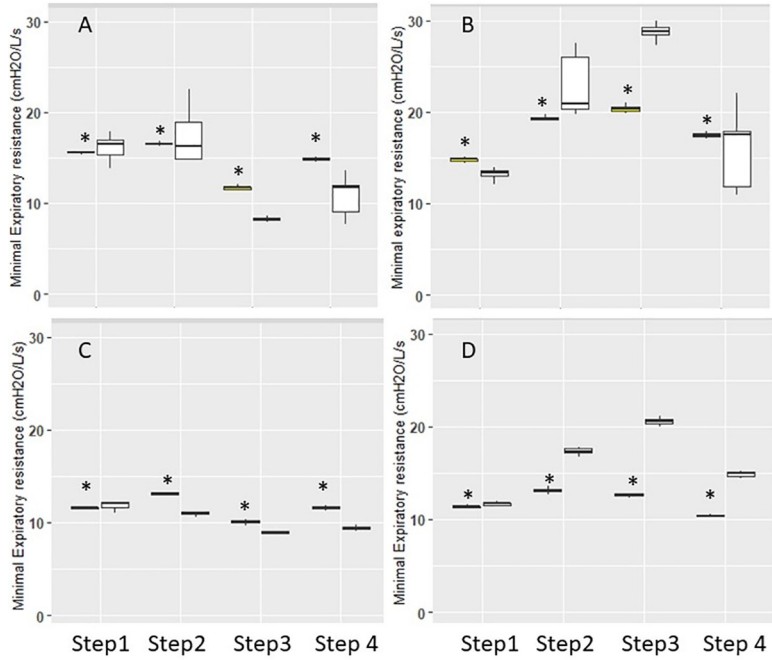

**Fig 4. Expiratory resistance.** Box-and-Whisker plots of minimal expiratory resistance in double (panel A and B) and triple circuits (panel C and D) with Elisée 350 (yellow) and V500 (white) ventilators during the different steps. Panels A and C are volume control and B and D pressure control. Whiskers denote median ± 1.58 x IQR x √3, where IQR is the interquartile range. *P<0.05 vs. V500.

## Discussion

We found that: 1) the target $V_T$ was achieved by the tested ventilators in volume and pressure control when they faced two symmetrical lungs, 2) asymmetrical C-R changed $V_T$ distribution between test-lungs, 3) the risk for pendelluft air from one test-lung to the other was related to the difference in lung time constants and also to the ventilator, 4) the performance of the two ventilators was close.

The shortage in ventilators to support COVID-19 patients in acute respiratory failure results from the imbalance between an acute enormous demand and a limited supply. The response of the healthcare system was a dramatic increase in the number of ICU beds in a very short period but the number of ventilators available was an issue. Trying to share ventilators is typical behavior of the "we have to do something" concept [17] to provide a fair allocation of resources [18–20] in the COVID-19 pandemic.

Started before the current COVID-19 pandemic [3, 21], it was stressed that simultaneous ventilation cannot support its use in mass causality because $V_T$ was too much variable across C-R conditions and largely dependent on changes in compliance [3]. Since then, the current COVID-19 pandemic prompted additional bench studies to extend these previous results and proposed solutions to try to overwhelm some related issues [9, 12, 22–34] (Table 5).

To make it clear from the very onset, we would like to state that simultaneous ventilation is a not validated experimental treatment and, hence should not be used. However, as a last resort in extreme cases it can be an option and has been actually carried out in real patients as already mentioned. Before discussing our results, the consideration of the rationale of our measurements is required. We assessed $V_T$ because it is the final goal of the ventilation, either spontaneous or assisted by a ventilator and choose a value that matched the 6 mL/kg average predicted body weight in ARDS patients in the Lung Safe study [25]. The PEEP of 15 cmH$_2$O

**Table 4. Sex, anthropometric characteristics, ventilator settings and respiratory mechanics of 20 patients with COVID-19 related ARDS.**

| Patient number | Sex | Weight (kg) | Height (cm) | BMI (kg/m²) | $V_T$ (mL) | $V_T$ (mL/kg PBW) | Respiratory rate (Breaths/min) | PEEP (cmH₂O) | FiO₂ (%) | Plateau pressure (cmH₂O) | Driving Pressure (cmH₂O) | Compliance (mL/cmH₂O) | Resistance (cmH₂O/L/s) |
|---|---|---|---|---|---|---|---|---|---|---|---|---|---|
| 1 | M | 85 | 174 | 28 | 420 | 6 | 25 | 7 | 100 | 18 | 11 | 39.4 | 7.3 |
| 2 | M | 111 | 176 | 36 | 388 | 5 | 21 | 12 | 100 | 24 | 12 | 31.7 | 10.3 |
| 3 | M | 112 | 184 | 33 | 440 | 6 | 32 | 14 | 100 | 27 | 13 | 34.0 | 10.4 |
| 4 | M | 98 | 177 | 31 | 389 | 5 | 20 | 10 | 100 | 21 | 11 | 34.0 | 15.2 |
| 5 | M | 82 | 175 | 27 | 382 | 5 | 24 | 10 | 80 | 24 | 14 | 26.5 | 10.7 |
| 6 | F | 75 | 168 | 27 | 322 | 5 | 28 | 8 | 70 | 20 | 12 | 26.0 | 12.3 |
| 7 | M | 114 | 185 | 33 | 447 | 6 | 20 | 10 | 85 | 24 | 14 | 32.8 | 16.2 |
| 8 | M | 93 | 176 | 30 | 440 | 6 | 26 | 15 | 100 | 23 | 8 | 54.7 | 9.1 |
| 9 | M | 68 | 165 | 25 | 420 | 7 | 25 | 10 | 100 | 19 | 9 | 46.0 | 9.5 |
| 10 | M | 82 | 171 | 28 | 450 | 7 | 20 | 14 | 85 | 26 | 12 | 38.0 | 10.7 |
| 11 | M | 82 | 168 | 29 | 420 | 7 | 22 | 12 | 100 | 26 | 14 | 30.3 | 9.8 |
| 12 | M | 72 | 172 | 24 | 400 | 6 | 24 | 10 | 80 | 20 | 10 | 38.3 | 8.3 |
| 13 | M | 112 | 180 | 35 | 347 | 5 | 18 | 15 | 100 | 29 | 14 | 24.9 | 14.1 |
| 14 | M | 120 | 169 | 42 | 336 | 5 | 23 | 14 | 50 | 22 | 8 | 41.0 | 11.3 |
| 15 | M | 99 | 180 | 31 | 416 | 6 | 27 | 10 | 100 | 19 | 9 | 47.4 | 13.2 |
| 16 | M | 92 | 172 | 31 | 410 | 6 | 20 | 14 | 100 | 26 | 12 | 35.2 | 8.4 |
| 17 | F | 68 | 156 | 28 | 292 | 6 | 28 | 12 | 100 | 27 | 15 | 19.4 | 15.2 |
| 18 | M | 70 | 176 | 23 | 303 | 4 | 24 | 10 | 100 | 32 | 22 | 13.8 | 14.7 |
| 19 | M | 99 | 179 | 31 | 462 | 6 | 26 | 14 | 100 | 28 | 14 | 34.0 | 11.2 |
| 20 | M | 71 | 168 | 25 | 345 | 5 | 17 | 12 | 75 | 18 | 6 | 55.6 | 10.5 |

BMI = body mass index, VT = tidal volume, PBW = predicted body weight, PEEP = positive end-expiratory pressure, FIO₂ = oxygen fraction in air.

**Table 5. Summary of the bench studies on shared ventilation between two test-lungs.**

| Author | Ventilator | Lung model | Data logger | VT target for ventilator (mL) | PEEP (cmH₂O) | RC test-lung 1 | RC test-lung 2 | $V_T$ Volume control (mL) test-lungs 1/2 | $V_T$ Pressure control (mL) test-lungs 1/2 |
|---|---|---|---|---|---|---|---|---|---|
| Chatburn | Servo-i | ASL 5000 | ASL 5000 | 800 | 15 | 10–45 | 10–45 | 396/392 | 421/417 |
| | | | | | | 10–45 | 10–20 | 406/293 | 500/272 |
| | | | | | | 10–50 | 10–20 | 467/267 | 485/250 |
| | | | | | | 10–45 | 15–45 | 440/349 | 457/369 |
| | | | | | | 10–45 | 30–45 | 499/216 | 493/241 |
| | | | | | | 10–20 | 25–60 | 406/382 | 376/412 |
| Epstein | Servo air | NA | PF300 | 1000 | 8 | NA-37 | NA-24 | 473/314 | 475/333 |
| Herrmann | Servo 300 | DemoLung in one patient | Florian Monitor | 1000 | 5 | 20–50 | 20–50 | 438/482 | 443/475 |
| | | | | | | 20–50 | 20–22 | 592/337 | 460/196 |
| | | Parabolic resistor and anesthesia reservoir bag in the other | | | | 20–50 | 20–7.5 | 735/159 | NA/NA |
| | | | | | | 20–50 | 20–35 | NA/NA | 451/332 |
| Han | PB840 | QuickLung | FlexMed Gr | 800 | 5 | NA-50 | NA-10 | NA/NA | 390/262 |
| Tonetti | Siaretron 4000 T (turbine) | Michigan Instruments 5601 | NA | 960 | 15 | 5–50 | 5–50 | NA/NA | 470/470 |
| | | | | | | 5–40 | 5–60 | NA/NA | 340/540 |
| | | | | | | 5–50 | 20–50 | NA/NA | 480/400 |

$V_T$ = tidal volume, PEEP = positive end-expiratory pressure, R = resistance, C = compliance, NA = not available.

was chosen to stretch the ventilator. The values of compliance selected were those allowed by the test-lung that we used. In COVID-19 related ARDS, compliance spread out from 20 to 90 ml/cmH$_2$O, with 50 ml/cmH$_2$O the most frequent value [26]. Further studies reported values of compliance in the range of 26–65 mL/cmH$_2$O [27–29]. In step 1, a similar C50R5 was applied to both test-lungs, to replicate type L COVID-19-related acute respiratory distress syndrome (ARDS) [30]. The rationale for measuring expiratory resistance was that hindered passive expiration can result in intrinsic PEEP (and dynamic hyperinflation) with deleterious consequence to the patients, such as hemodynamic impairment. Expiration is also frequently neglected in ventilator assessment. We used a single value of expiratory resistance to summarize as simply as possible the information of the complex process of the active expiratory valve functioning with a resistance that is continuously changing. Of note, we cannot control this issue on bench studies. We used the minimal expiratory resistance because this value is the closest level of expiratory resistance related to the set PEEP, and because it informs about the most likely speed at which the set PEEP returns [16]. We compared volume and pressure control ventilation because these are the most frequently used modes in the early phase of invasive ventilation when it is passive [31] and also each of these modes can be the single one at this stage in some countries, excluding the other. Finally, we added a third test-lung to take advantage of the bench condition to simulate an extreme condition.

We found that pressure control should be the mode of choice because it preserves V$_T$ in the least injured test-lung while volume control sets the healthier test-lung to overdistension and the worst test-lung to hypoventilation. This is, however, true only if the two test-lungs do not worsen differently in the same time. Test-lung B always received lower V$_T$ than expected at step 3 that may cause hypoventilation and CO$_2$ retention in the real patients. By comparison to the step 1 the introduction of asymmetry in pressure control does not compromise the target V$_T$ of the test-lung whose R-C set is not too much modified (test-lung A steps 2 and 4 in the 2 test-lungs design, test-lung C over all steps in the design with 3 test-lungs). Technical innovations have been proposed to individualize ventilator settings in each test-lung/patient, such as set inspiratory pressure, PEEP, and F$_I$O$_2$. These innovations include a one-way flow control valve at inspiratory and expiratory limbs in each test-lung/patient [13], a fixed pressure resistor regulator added at the inspiratory limb [32], a variable flow restrictor at the inspiratory limb, and a one-way valve at the expiratory limb [33], a flow restrictor on-way valve at the outlet of the ventilator [10], and bag-in-the box [23]. It should be noted that even though some of the interventions described above have been tested in a few patients [13], the experience is limited, they are complex to use and may generate further severe problems, as in case of an acute change in respiratory mechanics or gas exchange in one or two patients if the staff is not well trained.

The present study brings up new findings by testing two ventilators of different categories and measured pendelluft air and expiratory resistance of the ventilator valve. Even though most of the differences in V$_T$ between ventilators were statistically significant the clinical significance of them was irrelevant, meaning that the performance of the lower-level ventilator was close to that of the ICU ventilator. Therefore, the present findings suggest that the shortage of ICU ventilators can be overcome by using safely lower-level ventilators like the one we tested.

The amount of pendelluft air from one test-lung to the other is another issue of simultaneous ventilation. We quantified this in the present study both during inspiration and expiration (Fig 2). Even though the pendelluft air is lower than the anatomical dead space (including the endotracheal tube volume) and arises at the end of the inspiration of the test-lung/patient it would not induce CO$_2$ retention but sets the test-lung/patient at an increased risk of cross-infection. Pendelluft air can be explained by a difference of "plateau pressure" between the two

test-lungs as depicted in Fig 2. We expected that pressure control should prevent pendelluft from the following considerations. In terms of pressurization pressure control mode tends to favor fast and high pressurization finishing with a plateau while the volume control mode with its volume target tends to use less fast pressurization with a continuous increase of the pressure. In other words, we can expect that the pressure control mode is, at the end of the inspiration phase closer to an equilibrium situation for the plateau pressures than the volume control mode. The difference in pendelluft observed between the two ventilators is not simple to explain. Nevertheless, the difference observed in minimal expiratory resistance may also suggest that the opening speed of expiratory valves may be different between the ventilators. A delay to open the valve should favor gas recirculation between the two lungs at the start of the expiratory phase.

Some limitations should be mentioned that: 1) only two brands of ventilator were used and 2) the findings from this bench study might not be able to apply for patients, in particular those with spontaneous breathing. However, in the COVID-19 setting such an approach should be used in patients under deep sedation and neuromuscular blockade, early in the course of mechanical ventilation and for a few hours.

## Clinical implications

Even though experimental and not validated, simultaneous ventilation has actually been used in patients, as mentioned above. Beitler et al proposed four criteria that paired patients should comply with to be eligible for the technique [11]: same respiratory pathogen, 0–6 cmH$_2$O driving pressure difference, 0–8 breaths/min difference in respiratory rate and 0–5 cmH$_2$O difference in PEEP, with the difference in driving pressure the most important criterion to minimize. In a clinical scenario of shortage in ventilator, as can be seen from our sample of COVID-19 related ARDS, several patients could be enrolled in pairs to be treated accordingly, at least at an early stage after start of invasive mechanical ventilation (Table 4). The clinical implication of present findings is that if one ventilator is dedicated to two or three test-lungs, pressure control mode should be preferred and V$_T$ and end-tidal CO$_2$ of each patient should be closely monitored. Those patients should be managed by skilled caregivers, which would require an extensive education schedule in particular in those ICUs where pressure control is not routinely used. They would be able to understand and alert the clinical team of important changes in each of these patients.

An efficient filter upstream of the expiratory valve is mandatory. More complex implementations mentioned above have been achieved but may also generate problems. The pendelluft air supposes not only some difference in pressure between the two test-lungs but also an expiratory circuit common to the two test-lungs (including the expiratory valve of the ventilator) sufficiently resistive. We recently found that expiratory resistance differs between ICU ventilators [16]. Therefore, adding a poorly resistive valve at the beginning of the expiratory circuit should deemphasize the phenomenon of pendelluft air Nevertheless, the presence of such a valve could affect the capacities of the ventilator to maintain the PEEP.

Preclinical studies should be considered to increase our experience and knowledge in simultaneous ventilation in the dawn of the second wave of COVID-19 pandemic [34].

## Conclusion

The lower-level ventilator performed closely to the ICU ventilator. Due to dependence of V$_T$ to C pressure control should be used to maintain adequate V$_T$ at least in one test-lung when compliance and/or R change abruptly and monitoring of V$_T$ should be done carefully.

## Acknowledgments

The authors would like to thank Michelin France for providing us with the flow-splitter, Maud Grammatica MD and Sylvain Guibert for their help to get the flow-splitter.

## Author Contributions

**Conceptualization:** Claude Guérin, Erwan L'Her, Bruno Louis, Laurent Argaud.

**Data curation:** Claude Guérin, Neven Stevic, Florian Degivry, Bruno Louis, Laurent Argaud.

**Formal analysis:** Claude Guérin, Bruno Louis.

**Investigation:** Claude Guérin, Florian Degivry, Laurent Argaud.

**Methodology:** Claude Guérin, Erwan L'Her, Bruno Louis.

**Resources:** Laurent Argaud.

**Software:** Bruno Louis.

**Supervision:** Bruno Louis, Laurent Argaud.

**Validation:** Claude Guérin, Martin Cour, Neven Stevic, Florian Degivry, Bruno Louis.

**Visualization:** Claude Guérin, Martin Cour.

**Writing – original draft:** Claude Guérin.

**Writing – review & editing:** Claude Guérin, Martin Cour, Neven Stevic, Florian Degivry, Erwan L'Her, Bruno Louis, Laurent Argaud.

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
