## [Decision Letter · Decision Letter 0]

24 Nov 2020

PONE-D-20-34127

Sharing ventilators in the Covid-19 pandemics. A bench study

PLOS ONE

Dear Dr. Guérin,

Thank you for submitting your manuscript to PLOS ONE. After careful consideration, we feel that it has merit but does not fully meet PLOS ONE’s publication criteria as it currently stands. Therefore, we invite you to submit a revised version of the manuscript that addresses the points raised during the review process.

Both reviewers raised several concerns regarding your work. Particularly, the evaluation from the reviewer 1 was very critical. The authors need to effectively respond to their comments in their revision.

We look forward to receiving your revised manuscript.

Kind regards,

Yu Ru Kou, PhD

Academic Editor

PLOS ONE

Journal Requirements:

2. Please note that PLOS does not permit references to “data not shown.” Authors should provide the relevant data within the manuscript, the Supporting Information files, or in a public repository. If the data are not a core part of the research study being presented, we ask that authors remove any references to these data.

3. We note that Figure 1 in your submission contain copyrighted images. All PLOS content is published under the Creative Commons Attribution License (CC BY 4.0), which means that the manuscript, images, and Supporting Information files will be freely available online, and any third party is permitted to access, download, copy, distribute, and use these materials in any way, even commercially, with proper attribution. For more information, see our copyright guidelines: http://journals.plos.org/plosone/s/licenses-and-copyright.

3.1.         You may seek permission from the original copyright holder of Figure 1 to publish the content specifically under the CC BY 4.0 license.

3.2.    If you are unable to obtain permission from the original copyright holder to publish these figures under the CC BY 4.0 license or if the copyright holder’s requirements are incompatible with the CC BY 4.0 license, please either i) remove the figure or ii) supply a replacement figure that complies with the CC BY 4.0 license. Please check copyright information on all replacement figures and update the figure caption with source information. If applicable, please specify in the figure caption text when a figure is similar but not identical to the original image and is therefore for illustrative purposes only.

4. Please ensure you have discussed any potential limitations of your study in the Discussion.

5. During your revisions, please confirm whether the wording in the title is correct and update it in the manuscript file and online submission information if needed. Specifically, you may wish to change plural "pandemics" to singular "pandemic".

Reviewers' comments:

Reviewer's Responses to Questions

**Comments to the Author**

1. Is the manuscript technically sound, and do the data support the conclusions?

Reviewer #1: No

Reviewer #2: Yes

2. Has the statistical analysis been performed appropriately and rigorously? 

Reviewer #1: No

Reviewer #2: Yes

3. Have the authors made all data underlying the findings in their manuscript fully available?

Reviewer #1: No

Reviewer #2: Yes

4. Is the manuscript presented in an intelligible fashion and written in standard English?

Reviewer #1: No

Reviewer #2: Yes

5. Review Comments to the Author

Reviewer #1: Unfortunately I cannot suggest the manuscript “Sharing ventilators in the Covid-19 pandemics. A bench study” for publication in PLOS ONE in its present form

Multiplex ventilation appeared important in the light of the recent shortage of mechanical ventilators to maintain life of Covid 19 patients in the ICU. However, after several clinical studies and attempts to overcome some technical challenges (as correctly pointed in the present manuscript) the multiplex ventilation remains an experimental procedure.

In this regard the present a manuscript aimed to compare the ‘performance’ of two mechanical ventilators.

My major concern

1. Overall in the text the authors interpreted the results obtained form test lungs as results obtained from patients.

2. ‘We found that pressure control should be the mode of choice because it preserves VT in the least injured lung while volume control sets the healthier lung to overdistension and the worst lung to hypoventilation.`

such suggestion should be taken with a huge grain of salt. To my understanding authors suggest the using of PCV in case of multiplex ventilation because there will be a chance for the healthier patient to survive? I think the correct suggestion in this case should say that multiplex ventilation should be avoided?

3. The rational to measure Vt and minimal expiratory resistance in term to asses ventilator´s performance is not clear. In this respect the need of experimental set with 3 test lungs and the comparison of two ventilation modes (VCV and PCV) were not explained.

4. Regarding Vt: ‘P<0.001 between ventilators at each step for each volume’

if so how did the authors conclude that ‘The lower-level ventilator performed closely to the ICU-dedicated ventilator’

The paper required extensive language check (particularly, punctuation and logic but in some cases also the correct word/term usage)

Methods and results sections required improvement/restructuring. Some rational is discussed in the methods section!?

5. how many repeated measurements where performed for each ‘step’?

6. I feel that the term ‘rebreathed volume’ is not entirely correct. In the presented model air is transported from one lung (or compartment) to another based on differences in the respiratory mechanics of the lungs (compartments) this is referred to as pendulum air (doi: 10.1007/BF02469481). Very important! According to the presented in fig.2 results this transport takes place before the beginning of expiration? (Blue aria ends with the begin of expiration)?

7. Did you test your data for homogeneity of variance and normal distribution? What was the rational to use a non-parametric test? Was this test used to compare only the ‘C-R’ data?

8. ‘Bars are median and quartiles omitted for clarity’ Please proved the omitted quartiles or show +/-SD.

9. The results description and presentation are not uniform and not helpful for the reader.

-Fig.1 Describes only one of the two experimental sets (it will be helpful to use different colors or arrows for Inspiration and Expiration).

-Fig.2 Appears like a presentation slide. 1. Please provide the individual pressure curves (as shown for the flow curves). 2. PEEP is definitively above 15? 3. Paw of 40 (or plateau 35 cm H2O) is definitively injurious? Are data presented there from first set (2 lungs model) and from which ‘Step’ ?

-Fig 3 do not include SD or any indication of variety of experimental data.

- if correctly understood you have measured in the expiration… 2s (400 data points) but then discarded the long PEEP phase so final you have calculated the resistance changes during expiration! what is the rational in this case to take a single value and particularly the minimum resistance for further comparisons?

10. to what extend did the HEPA filters contributed to the minimal resistance ?

11. It is interesting … what is the base for the observed extreme variation of the minimal R in case of mechanical ventilation with V500 in the 2 lung model (fig4 panels A and B)?

12. It is also a bit odd that two groups with very similar or even identical medians and overlapping data points are showing significant differences (e.g fig. 4 A step 2 and B step 4)?

Minor notes

Title

- ‘Sharing’ do not necessary implies a simultaneous MV application?

Abstract

- the use of some words is ambiguous (e.g dedicated)

- The description of the experimental groups should be improved to facilitate reader to appreciate the presented study results (eg for AC50R5/B C50R5) Vt was 0.381/0.387)

Introduction

-Please, describe the meaning of ‘ventilator multipliers’ it was not to find in the mention citation!

Methods

- ‘Drager , Lubeck’.Consider the use of Draeger, Luebeck

- ‘lung test’. you probably mean test lung

- ‘and 20 cmH2O/L/s ± was added’. The 20 cm H2O is probably the resistance (R)? what does the ± stands for?

- ‘contrasted time constants’ 1. Time constants (R*C product) are not shown in Table 1 and 2. to which timepoint (tau 1, 2)?

- ‘PEEP to 15 cmH2O, a value chosen to stretch the ventilator’. Does PEEP of 15 stretch one or both ventilators? and what is the effect of ventilator stretching ?

- Table 2 is a bit redundant. FiO2, Vt, and PEEP are described in the text.

- Table 2 Driving pressure is not a ventilator setting?

-Table 2 first-third quartiles are mentioned but not shown?

-‘ safety guard`’ or safety guidelines?

-‘ In the second design, a third lung with a fixed C20-R20’ or C was set to 10 as mentioned earlier?

- please clarify the term ‘instantaneous expiratory resistance ‘?

-‘ On each breath, the instantaneous expiratory resistance was determined as the ratio of the pressure drop between Paw and atmosphere’ Atmosphere or as used in the study PEEP of 15 cm H2O?

-‘Fig 4. Expiratory resistance. Box-and-Whisker plots’ I see only box plots with whiskers showing something not defined by the authors! - what are the whiskers in the box plots showing?

Discussion

- how can the FiO2 contribute to homogenous gas distribution? how can be PEEP and FiO2 set individually in a tow lung model ??

Reviewer #2: Guerin and colleagues performed an excellent bench study regarding the ventilator sharing. Two types of ventilator including high performance ICU ventilator (V500) and lower level ventilator (Elysee 350) were evaluated using the test lung with different respiratory mechanics. The flow splitter was used to seperate the flow delivery to two and three test lungs. They showed that the lower-level ventilator performed closely to the ICU-dedicated ventilator in terms of the delivered VT, rebreated volume and minimal expiratory resistance. The results of this study would be useful when we need to share the ventilator to two or three patients during COVID-19 pandemic because of ventilator shortage. However, I have comments and suggestions that need to be clarify as follows:

1) Major comments

Methods

• Did the authors calibrate the ventilator and check the circuit leakage before starting the experiments? It should be mentioned in the methods.

• The measured VT might be different from the delivered VT by the ventilator because of the circuit compliance and gas compressibility in the circuit. Did the authors concern about these factors?

• The authors only explained how to desire the compliance using the reported values of respiratory system compliance in patients with COVID19 but they did not mention how to desire the resistance (I was wondering why they set quite high resistance in the most experiments instead of using normal resistance if they want to reflect patients with COVID19 pneumonia/ARDS)?.

• How many breaths were used for calculating all variables? It should be described in methods.

Results

• I would suggest to present the same unit of VT and rebreathed volume (L or mL).

Discussion

• From their experiments, the authors suggested that PCV was better than VCV in terms of lung protective ventilation for less injured lung and the worst lung. However, I’m not sure that this statement is true because patient B always received lower VT than expected in particular when we look at the result from step 3 that demonstrated very low VT in both lungs (less than 200 mL) and it may cause hypoventilation and CO2 retention in the real patients.

• How can they conclude that adjusting inspiratory pressure during PCV in step 2-4 will increase VT in the preserved lung without the experiment to confirm their hypothesis.

• Some limitations should be mentioned that 1) only two brands of ventilator were used and 2) the findings from this bench study might not be able to apply for patients with spontaneous breathing, etc.

2) Minor comments

• Page 2: Please insert “vs.” for 0.416/0.185/0.322/0.234L

• Page 5: Please remove “±” after 20 cmH2O/L/s.

• Page 11: Please check the value of VT at step 2 for patient B with V500 (the median value should not be 0.70 L).

6. PLOS authors have the option to publish the peer review history of their article (what does this mean?). If published, this will include your full peer review and any attached files.

Reviewer #1: No

Reviewer #2: No

---

## [Author Response · Author response to Decision Letter 0]

23 Dec 2020

PONE-D-20-34127

Simultaneous ventilation in the Covid-19 pandemic. A bench study

 PLOS ONE

 Dear Dr. Guérin,

Thank you for submitting your manuscript to PLOS ONE. After careful consideration, we feel that it has merit but does not fully meet PLOS ONE’s publication criteria as it currently stands. Therefore, we invite you to submit a revised version of the manuscript that addresses the points raised during the review process.

Both reviewers raised several concerns regarding your work. Particularly, the evaluation from the reviewer 1 was very critical. The authors need to effectively respond to their comments in their revision.

 Please submit your revised manuscript by Jan 08 2021 11:59PM. If you will need more time than this to complete your revisions, please reply to this message or contact the journal office at plosone@plos.org. Please include the following items when submitting your revised manuscript:

•A rebuttal letter that responds to each point raised by the academic editor and reviewer(s). You should upload this letter as a separate file labeled 'Response to Reviewers'.

R. Done

•A marked-up copy of your manuscript that highlights changes made to the original version. You should upload this as a separate file labeled 'Revised Manuscript with Track Changes'.

R. Done

•An unmarked version of your revised paper without tracked changes. You should upload this as a separate file labeled 'Manuscript'.

R. Done

2. Please note that PLOS does not permit references to “data not shown.” Authors should provide the relevant data within the manuscript, the Supporting Information files, or in a public repository. If the data are not a core part of the research study being presented, we ask that authors remove any references to these data.

R. There is no longer “data not shown” in the revised version.

3. We note that Figure 1 in your submission contain copyrighted images. All PLOS content is published under the Creative Commons Attribution License (CC BY 4.0), which means that the manuscript, images, and Supporting Information files will be freely available online, and any third party is permitted to access, download, copy, distribute, and use these materials in any way, even commercially, with proper attribution. For more information, see our copyright guidelines: http://journals.plos.org/plosone/s/licenses-and-copyright.

R. We deleted the photographs with copyrighted images of some devices in the revised version. We replace these as drawn objects.

4. Please ensure you have discussed any potential limitations of your study in the Discussion.

R. We added some other limitations of our study in the discussion section of the revised version.

 5. During your revisions, please confirm whether the wording in the title is correct and update it in the manuscript file and online submission information if needed. Specifically, you may wish to change plural "pandemics" to singular "pandemic".

R. Thank you. The title has been changed as “Simultaneous ventilation in the COVID-19 pandemic. A bench study” in the revised version to take into account your remark as that of one reviewer.

Reviewers' comments:

 Reviewer's Responses to Questions

Author

 1. Is the manuscript technically sound, and do the data support the conclusions?

Reviewer #1: No

Reviewer #2: Yes

2. Has the statistical analysis been performed appropriately and rigorously? 

Reviewer #1: No

Reviewer #2: Yes

3. Have the authors made all data underlying the findings in their manuscript fully available?

Reviewer #1: No

Reviewer #2: Yes

4. Is the manuscript presented in an intelligible fashion and written in standard English?

Reviewer #1: No

Reviewer #2: Yes

5. Review Comments to the Author

Reviewer #1: Unfortunately I cannot suggest the manuscript “Sharing ventilators in the Covid-19 pandemics. A bench study” for publication in PLOS ONE in its present form

 Multiplex ventilation appeared important in the light of the recent shortage of mechanical ventilators to maintain life of Covid 19 patients in the ICU. However, after several clinical studies and attempts to overcome some technical challenges (as correctly pointed in the present manuscript) the multiplex ventilation remains an experimental procedure.

R. Thank you for this comment. We do agree with your last sentence that multiplex ventilation is experimental and should not be used. Nevertheless, it has been carried out in some patients in reports we quoted in our first version. 

 In this regard the present a manuscript aimed to compare the ‘performance’ of two mechanical ventilators.

 My major concern

 Overall in the text the authors interpreted the results obtained form test lungs as results obtained from patients.

R. Thank you for this comment. You are correct. In the revised version we replaced “patients” by “test-lungs” any time it is required. In some places the word “patient” was kept when we thought it was more appropriate.

 2. ‘We found that pressure control should be the mode of choice because it preserves VT in the least injured lung while volume control sets the healthier lung to overdistension and the worst lung to hypoventilation.`

 such suggestion should be taken with a huge grain of salt. To my understanding authors suggest the using of PCV in case of multiplex ventilation because there will be a chance for the healthier patient to survive? I think the correct suggestion in this case should say that multiplex ventilation should be avoided?

R. Thank you for this comment. We do agree with you that simultaneous ventilation is experimental and should not be used as recommended by experts and professionals organization. However, it has been used and reported and we quoted these papers. What we meant is if simultaneous ventilation should be carried as a last resort for extreme cases then pressure control should preferred. We mentioned this statement at the beginning of the discussion in the revised version.

 The rational to measure Vt and minimal expiratory resistance in term to asses ventilator´s performance is not clear. In this respect the need of experimental set with 3 test lungs and the comparison of two ventilation modes (VCV and PCV) were not explained.

R. Thank you for this comment. In the revised version we added this rationale we placed in the discussion to be in line with another comment you have on the rationale. What we added is: “We assessed VT because it is the final goal of the ventilation, either spontaneous or assisted by a ventilator. The rationale for measuring expiratory resistance was that expiration being a passive process in the setting presently explored, intrinsic PEEP and dynamic hyperinflation may result when it is hindered with deleterious consequence, such as hemodynamic impairment. It is also frequently neglected in ventilator assessment. We compared volume and pressure control ventilation because these are the most frequently used modes in the early phase of invasive ventilation when it is passive (Esteban AJRCCM 2013) and also each of these can be the single one used at this stage in some countries, excluding the other. Finally, we added a third test-lung to take advantage of the bench condition to simulate an extreme condition”. 

 4. Regarding Vt: ‘P<0.001 between ventilators at each step for each volume’ if so how did the authors conclude that ‘The lower-level ventilator performed closely to the ICU-dedicated ventilator’

R. Because even though it is statistically significant it may be not clinically significant. Due to the bench condition, the reproducibility of the measurements over a substantial number of repeated measures makes that small differences are statistically significant even though they are not clinically sound.

The paper required extensive language check (particularly, punctuation and logic but in some cases also the correct word/term usage)

R. The revised version has been checked carefully for language.

 Methods and results sections required improvement/restructuring. Some rational is discussed in the methods section!?

R. Thank you for this comment. In the revised version we moved the rationale for VT, PEEP and compliance to the discussion section.

 5. how many repeated measurements where performed for each ‘step’?

R. Twenty. Mentioned in the revised version. Thank you.

6. I feel that the term ‘rebreathed volume’ is not entirely correct. In the presented model air is transported from one lung (or compartment) to another based on differences in the respiratory mechanics of the lungs (compartments) this is referred to as pendulum air (doi: 10.1007/BF02469481). Very important! 

R. Thank you for this important comment. We fully agree with you. In the revised version rebreathed volume has been replaced by “pendelluft air” throughout text and figures. We adopted this wording rather than pendulum air because it is more commonly used. Should you want us to use pendulum air we will do it.

According to the presented in fig.2 results this transport takes place before the beginning of expiration? (Blue aria ends with the begin of expiration)?

R. Thank you for this comment. Actually, in figure 2 pendelluft air takes place at the beginning of the expiration of the VENTILATOR but BEFORE expiration for the test-lung A and AFTER onset of expiration for the test-lung B. The blue area ends after the onset of the expiration of the test-lung B.

7. Did you test your data for homogeneity of variance and normal distribution? What was the rational to use a non-parametric test? Was this test used to compare only the ‘C-R’ data?

R. Thank you for this comment. Normal distribution and homogeneity of variables were assessed and depending on these results parametric or nonparametric test was chosen. 

 8. ‘Bars are median and quartiles omitted for clarity’ Please proved the omitted quartiles or show +/-SD.

R. In the revised version the figure 3 now shows a metric of dispersion of the values. Since we show median values and because it is awkward to show first and third quartiles simultaneously we present the interquartile range, which is the difference between the third to the first quartile. Therefore, as it would have been the case with SD there is a single bar for that. 

 9. The results description and presentation are not uniform and not helpful for the reader.

 -Fig.1 Describes only one of the two experimental sets (it will be helpful to use different colors or arrows for Inspiration and Expiration).

R. Thank you. In the revised version, a new panel has been added to this figure showing the three test-lungs.

-Fig.2 Appears like a presentation slide. 

R. In the revised version the figure 2 has been deeply changed and matches the requirement for a more classic illustration. Thank you.

1. Please provide the individual pressure curves (as shown for the flow curves). 

R. Thank you. Done in the revised version.

2. PEEP is definitively above 15? 

R. Yes. It is true. The PEEP was set to 15 but as measured it was slightly above this value.

3. Paw of 40 (or plateau 35 cm H2O) is definitively injurious? 

R. Yes. In this bench experiment we did not take of this.

Are data presented there from first set (2 lungs model) and from which ‘Step’ ?

R. Thank you for this comment. It was step 2 for the V500 ventilator. Mentioned in the revised version.

-Fig 3 do not include SD or any indication of variety of experimental data.

R. Please see our answer to your comment above: In the revised version the figure 3 now shows a metric of dispersion of the values. Since we show median values and because it is awkward to show first and third quartiles simultaneously we present the interquartile range, which is the difference between the third to the first quartile. Therefore, as it would have been the case with SD there is a single bar for that.

- if correctly understood you have measured in the expiration… 2s (400 data points) but then discarded the long PEEP phase so final you have calculated the resistance changes during expiration! what is the rational in this case to take a single value and particularly the minimum resistance for further comparisons?

R. Thank you for this comment. We used a single value of expiratory resistance to summarize as simply as possible the information of the complex process of the active expiratory valve functioning with a resistance that is continuously changing, a process we cannot control. We used the minimal expiratory resistance because this value is the closest level of expiratory resistance related to the set PEEP and informs about the most likely speed at which the set PEEP is going to return based on a recent study we reported (Pinede et al quoted). We took advantage of your comment to add these information in the part of the discussion section where the rationale is discussed in the revised version. Thank you.

 10. to what extend did the HEPA filters contributed to the minimal resistance ?

R. We measured the resistance of the HEPA filter and found 1.8 cmH2O/L/s, which is in line with the manufacturer specification.

 11. It is interesting … what is the base for the observed extreme variation of the minimal R in case of mechanical ventilation with V500 in the 2 lung model (fig4 panels A and B)?

R. Thank you. This variation depends on the continuous change in the position of the expiratory valve which under the control of a specific algorithm, which is trade secret. 

 12. It is also a bit odd that two groups with very similar or even identical medians and overlapping data points are showing significant differences (e.g fig. 4 A step 2 and B step 4)?

R. Yes but this is the statistical result we got and should be explained by the variability in one group and the reproducibility of the results in the other group. As mentioned above statistical difference may be far from the clinical difference.

 Minor notes

 Title

- ‘Sharing’ do not necessary implies a simultaneous MV application?

R. Thank you. According to your comment we changed sharing ventilators by simultaneous ventilation throughout in the revised version.

 Abstract

 - the use of some words is ambiguous (e.g dedicated)

R. We deleted “dedicated” throughout in the revised version.

 - The description of the experimental groups should be improved to facilitate reader to appreciate the presented study results (eg for AC50R5/B C50R5) Vt was 0.381/0.387)

R. Thank you. This has been fixed in the revised version.

 Introduction

-Please, describe the meaning of ‘ventilator multipliers’ it was not to find in the mention citation!

R. Thank you. Reference added in the revised version.

 Methods

- ‘Drager , Lubeck’.Consider the use of Draeger, Luebeck

R. Fixed in the revised version.

- ‘lung test’. you probably mean test lung

R. Yes. Thank you. Fixed in the revised version.

- ‘and 20 cmH2O/L/s ± was added’. The 20 cm H2O is probably the resistance (R)? what does the ± stands for?

R. Fixed in the revised version (± removed).

- ‘contrasted time constants’ 1. Time constants (R*C product) are not shown in Table 1 and 2. to which timepoint (tau 1, 2)?

R. Thank you. We added the time constant at each R-C combination in table 1 and table 3 (we understand that you meant table 3 and not table 2) in the revised version.

- ‘PEEP to 15 cmH2O, a value chosen to stretch the ventilator’. Does PEEP of 15 stretch one or both ventilators? 

R. PEEP 15 should stress both test-lungs.

and what is the effect of ventilator stretching ?

R. The effect is an increase in lung volume and hence a bigger challenge for the ventilator to maintain the set PEEP. We have to acknowledge that we did not test different PEEP levels in this study.

- Table 2 is a bit redundant. FiO2, Vt, and PEEP are described in the text.

R. Thank you. We removed the table 2 from the revised version and described the settings in the text.

 - Table 2 Driving pressure is not a ventilator setting?

R. Yes, you are correct driving pressure is a ventilator setting in pressure control ventilation. Table 2 has been removed from the revised version.

 -Table 2 first-third quartiles are mentioned but not shown?

R. You are right. This is a mistake. In the revised version this table 2 does not longer appear.

-‘ safety guard`’ or safety guidelines?

R. Yes safety guidelines. Fixed in the revised version.

-‘ In the second design, a third lung with a fixed C20-R20’ or C was set to 10 as mentioned earlier?

R. Thank you for picking-up this mistake. Fixed in the revised version (C10).

- please clarify the term ‘instantaneous expiratory resistance ‘?

R. It is the ratio of Paw to flow at every single sampled data point. Fixed in the revised version. Thank you.

-‘ On each breath, the instantaneous expiratory resistance was determined as the ratio of the pressure drop between Paw and atmosphere’ Atmosphere or as used in the study PEEP of 15 cm H2O?

R. Yes. You are correct it is Paw-PEEP difference divided by flow at every single sampled data point. Fixed in the R1. Thank you.

-‘Fig 4. Expiratory resistance. Box-and-Whisker plots’ I see only box plots with whiskers showing something not defined by the authors! - what are the whiskers in the box plots showing?

R. The whiskers showed median ± 1.58 x IQR x √3 With IQR the interquartile range. Fixed in the revised version. Thank you.

Discussion

 - how can the FiO2 contribute to homogenous gas distribution? 

R. If FIO2 does not contribute to homogenous gas distribution it can be seen as a marker of the quality of the ventilation distribution. In case of important pendelluft we can expect a decrease of FIO2 in a lung by comparison with the FIO2 delivered by the ventilator. Moreover different level of FIO2 can be wished for each patient.

how can be PEEP and FiO2 set individually in a tow lung model ??

R. This has been done for the PEEP for instance by using a bag-in-the box (Han et al CCE 2020, a reference we quoted in the previous version).

Reviewer #2: Guerin and colleagues performed an excellent bench study regarding the ventilator sharing. Two types of ventilator including high performance ICU ventilator (V500) and lower level ventilator (Elysee 350) were evaluated using the test lung with different respiratory mechanics. The flow splitter was used to seperate the flow delivery to two and three test lungs. They showed that the lower-level ventilator performed closely to the ICU-dedicated ventilator in terms of the delivered VT, rebreated volume and minimal expiratory resistance. The results of this study would be useful when we need to share the ventilator to two or three patients during COVID-19 pandemic because of ventilator shortage. 

R. Thank you very much for your appraisal of our work.

However, I have comments and suggestions that need to be clarify as follows:

 1) Major comments

 Methods

• Did the authors calibrate the ventilator and check the circuit leakage before starting the experiments? It should be mentioned in the methods.

R. Thank you for this comment. Yes we did. Mentioned in the methods section in the revised version.

• The measured VT might be different from the delivered VT by the ventilator because of the circuit compliance and gas compressibility in the circuit. Did the authors concern about these factors?

R. Thank you for this important comment. Yes, we are extremely concerned with that. Both ventilators compensate for the circuit compliance. The same circuit was used in both ventilators and for each experiment. The calibration process was done with this circuit attached to the two and three test-lungs designs. This was mentioned in the revised version.

• The authors only explained how to desire the compliance using the reported values of respiratory system compliance in patients with COVID19 but they did not mention how to desire the resistance (I was wondering why they set quite high resistance in the most experiments instead of using normal resistance if they want to reflect patients with COVID19 pneumonia/ARDS)?.

R. Thank you for this comment. Your point is well taken. Actually, the data about resistance in COVID-19 related ARDS is very limited (contrary to compliance). In the report by Bonny et al (Crit Care 2020,24:596) the resistance is extremely low (0.24 cmH2O/L/s) and may be a mistake. In ARDS non-related to COVID-19 resistance is also rarely reported at least in the most recent literature. According to Arnal et al (Resp Care 2018) values of resistance of 15-16 cmH2O/L/s are reported, which are not far from the highest value (20 cmH2O/L/s) we used in present study. Herrman et al (Shared Ventilation in the Era of COVID-19: A Theoretical Consideration of the Dangers and Potential Solutions. Resp Care 2020) also set out resistance of 20 cmH2O/L/s. Finally, in the most recent consecutive 20 ARDS patients with COVID-19 in our 26-bed medical ICU who were receiving invasive ventilation, resistance was between 10-17 cmH2O/L/s in 14 of them (new table 5 in the revised version, we get back to below). So, it is not completely irrelevant to set high value of resistance. It should also be mentioned that a bench study like ours present is well suited to test extreme situations.

• How many breaths were used for calculating all variables? It should be described in methods.

R. Thank you for this comment. Twenty. Mentioned in the methods section in the revised version.

 Results

• I would suggest to present the same unit of VT and rebreathed volume (L or mL).

R. Thank you for this comment. We used mL for VT in the revised version.

 Discussion

• From their experiments, the authors suggested that PCV was better than VCV in terms of lung protective ventilation for less injured lung and the worst lung. However, I’m not sure that this statement is true because patient B always received lower VT than expected in particular when we look at the result from step 3 that demonstrated very low VT in both lungs (less than 200 mL) and it may cause hypoventilation and CO2 retention in the real patients.

R. Thank you for this comment. You are correct that this is true if the two lungs do not worsen differently. We used your comment in the discussion in the revised version.

• How can they conclude that adjusting inspiratory pressure during PCV in step 2-4 will increase VT in the preserved lung without the experiment to confirm their hypothesis.

R. Thank you for this comment. You are correct. The sentence we removed from the revised version. 

• Some limitations should be mentioned that 1) only two brands of ventilator were used and 2) the findings from this bench study might not be able to apply for patients with spontaneous breathing, etc.

R. Thank you for this comment. We added these limitations in the revised version. 

 2) Minor comments

• Page 2: Please insert “vs.” for 0.416/0.185/0.322/0.234L

R. Thank you. This has been fixed in the revised version.

• Page 5: Please remove “±” after 20 cmH2O/L/s.

R. Thank you. This has been fixed in the revised version.

• Page 11: Please check the value of VT at step 2 for patient B with V500 (the median value should not be 0.70 L).

R. Thank you for picking up this mistake. It is 0.370 L. Fixed in the revised version.

6. PLOS authors have the option to publish the peer review history of their article (what does this mean?). If published, this will include your full peer review and any attached files.

 Do you want your identity to be public for this peer review? For information about this choice, including consent withdrawal, please see our Privacy Policy.

Reviewer #1: No

Reviewer #2: No

---

## [Decision Letter · Decision Letter 1]

4 Jan 2021

PONE-D-20-34127R1

Simultaneous ventilation in the Covid-19 pandemic. A bench study

PLOS ONE

Dear Dr. Guérin,

Thank you for submitting your manuscript to PLOS ONE. After careful consideration, we feel that it has merit but does not fully meet PLOS ONE’s publication criteria as it currently stands. Therefore, we invite you to submit a revised version of the manuscript that addresses the points raised during the review process.

One minor error to be corrected. Let us go one more round.

We look forward to receiving your revised manuscript.

Kind regards,

Yu Ru Kou, PhD

Academic Editor

PLOS ONE

Reviewers' comments:

Reviewer's Responses to Questions

**Comments to the Author**

1. If the authors have adequately addressed your comments raised in a previous round of review and you feel that this manuscript is now acceptable for publication, you may indicate that here to bypass the “Comments to the Author” section, enter your conflict of interest statement in the “Confidential to Editor” section, and submit your "Accept" recommendation.

Reviewer #1: All comments have been addressed

Reviewer #2: All comments have been addressed

2. Is the manuscript technically sound, and do the data support the conclusions?

Reviewer #1: Yes

Reviewer #2: Yes

3. Has the statistical analysis been performed appropriately and rigorously? 

Reviewer #1: Yes

Reviewer #2: Yes

4. Have the authors made all data underlying the findings in their manuscript fully available?

Reviewer #1: Yes

Reviewer #2: Yes

5. Is the manuscript presented in an intelligible fashion and written in standard English?

Reviewer #1: Yes

Reviewer #2: Yes

6. Review Comments to the Author

Reviewer #1: The Authors have improved the manuscript ‘Simultaneous ventilation in the Covid-19 pandemic. A bench study’ and addressed all issues to my satisfaction.

Reviewer #2: The authors have already addressed all of my comments and suggestions. I satisfy with the revised version. There is only one minor comment as follows:

- Page 7, 1st paragraph: please correct the unit of inspiratory flow setting from 60 "L/s" to be "L/min".

7. PLOS authors have the option to publish the peer review history of their article (what does this mean?). If published, this will include your full peer review and any attached files.

Reviewer #1: No

Reviewer #2: No

---

## [Author Response · Author response to Decision Letter 1]

4 Jan 2021

PONE-D-20-34127R1

 Simultaneous ventilation in the Covid-19 pandemic. A bench study

 PLOS ONE

 Dear Dr. Guérin,

 Thank you for submitting your manuscript to PLOS ONE. After careful consideration, we feel that it has merit but does not fully meet PLOS ONE’s publication criteria as it currently stands. Therefore, we invite you to submit a revised version of the manuscript that addresses the points raised during the review process.

One minor error to be corrected. Let us go one more round.

R. Thank you very much for your appraisal of our revised version. The error disclosed has been corrected in the R2.

Comments to the Author

Reviewer #1: The Authors have improved the manuscript ‘Simultaneous ventilation in the Covid-19 pandemic. A bench study’ and addressed all issues to my satisfaction.

R. Thank you very much. Your words are highly appreciated.

Reviewer #2: The authors have already addressed all of my comments and suggestions. I satisfy with the revised version. There is only one minor comment as follows:

 - Page 7, 1st paragraph: please correct the unit of inspiratory flow setting from 60 "L/s" to be "L/min".

R. Thank you very much. Your words are highly appreciated. Thank you for picking up this mistake. We have corrected 60 L/s to 60 L/min in the R2.

---

## [Editor Report · Decision Letter 2]

5 Jan 2021

Simultaneous ventilation in the Covid-19 pandemic. A bench study

PONE-D-20-34127R2

Dear Dr. Guérin,

We’re pleased to inform you that your manuscript has been judged scientifically suitable for publication and will be formally accepted for publication once it meets all outstanding technical requirements.

Kind regards,

Yu Ru Kou, PhD

Academic Editor

PLOS ONE
---

## [Editor Report · Acceptance letter]

6 Jan 2021

PONE-D-20-34127R2 

Simultaneous ventilation in the Covid-19 pandemic. A bench study 

Dear Dr. Guérin:

I'm pleased to inform you that your manuscript has been deemed suitable for publication in PLOS ONE. Congratulations! Your manuscript is now with our production department. 

Kind regards, 

on behalf of

Dr. Yu Ru Kou 

Academic Editor

PLOS ONE